# Prediction of Prostate Cancer Biochemical and Clinical Recurrence Is Improved by IHC-Assisted Grading Using Appl1, Sortilin and Syndecan-1

**DOI:** 10.3390/cancers15123215

**Published:** 2023-06-16

**Authors:** Jessica M. Logan, Ashley M. Hopkins, Carmela Martini, Alexandra Sorvina, Prerna Tewari, Sarita Prabhakaran, Chelsea Huzzell, Ian R. D. Johnson, Shane M. Hickey, Ben S.-Y. Ung, Joanna Lazniewska, Robert D. Brooks, Courtney R. Moore, Maria C. Caruso, Litsa Karageorgos, Cara M. Martin, Sharon O’Toole, Laura Bogue Edgerton, Mark P. Ward, Mark Bates, Stavros Selemidis, Adrian Esterman, Sheena Heffernan, Helen Keegan, Sarah Ní Mhaolcatha, Roisin O’Connor, Victoria Malone, Marguerite Carter, Katie Ryan, Andres Clarke, Nathan Brady, Sonja Klebe, Hemamali Samaratunga, Brett Delahunt, Michael J. Sorich, Kim Moretti, Lisa M. Butler, John J. O’Leary, Douglas A. Brooks

**Affiliations:** 1Clinical and Health Sciences, University of South Australia, Bradley Building, City West Campus, North Terrace, Adelaide, SA 5000, Australia; 2College of Medicine and Public Health, Flinders University, Flinders Drive, Bedford Park, Adelaide, SA 5042, Australia; 3Department of Histopathology, Trinity College Dublin, D02 PN40 Dublin, Ireland; 4Department of Anatomical Pathology, College of Medicine and Public Health, Flinders University, Adelaide, SA 5042, Australia; 5School of Health and Biomedical Sciences, STEM College, RMIT University, Bundoora, Melbourne, VIC 3001, Australia; 6Department of Pathology, The Coombe Women and Infants University Hospital, D08 XW7X Dublin, Ireland; 7Aquesta Uropathology, The University of Queensland, Brisbane, QLD 4072, Australia; 8Malaghan Institute of Medical Research, Wellington 6012, New Zealand; 9Discipline of Surgery, University of Adelaide, Adelaide, SA 5371, Australia; 10Allied Health and Human Performance, University of South Australia, Adelaide, SA 5005, Australia; 11Faculty of Medicine, Nursing and Health Sciences, Monash University, Melbourne, VIC 3800, Australia; 12South Australian ImmunoGENomics Cancer Institute and Freemasons Centre for Male Health and Wellbeing, University of Adelaide, Adelaide, SA 5005, Australia; 13Solid Tumour Program, Precision Cancer Medicine Theme, South Australian Health and Medical Research Institute, Adelaide, SA 5000, Australia

**Keywords:** biomarkers, diagnosis, prognosis, prostate cancer, clinical recurrence

## Abstract

**Simple Summary:**

A novel biomarker panel (Appl1, Sortilin and Syndecan-1) was demonstrated as a strong independent predictor for both clinical and biochemical recurrence outcomes, with a higher prediction performance than traditional grading. This suggests that panel-derived patient reclassifications improve risk stratification.

**Abstract:**

Gleason scoring is used within a five-tier risk stratification system to guide therapeutic decisions for patients with prostate cancer. This study aimed to compare the predictive performance of routine H&E or biomarker-assisted ISUP (International Society of Urological Pathology) grade grouping for assessing the risk of biochemical recurrence (BCR) and clinical recurrence (CR) in patients with prostate cancer. This retrospective study was an assessment of 114 men with prostate cancer who provided radical prostatectomy samples to the Australian Prostate Cancer Bioresource between 2006 and 2014. The prediction of CR was the primary outcome (median time to CR 79.8 months), and BCR was assessed as a secondary outcome (median time to BCR 41.7 months). The associations of (1) H&E ISUP grade groups and (2) modified ISUP grade groups informed by the Appl1, Sortilin and Syndecan-1 immunohistochemistry (IHC) labelling were modelled with BCR and CR using Cox proportional hazard approaches. IHC-assisted grading was more predictive than H&E for BCR (C-statistic 0.63 vs. 0.59) and CR (C-statistic 0.71 vs. 0.66). On adjusted analysis, IHC-assisted ISUP grading was independently associated with both outcome measures. IHC-assisted ISUP grading using the biomarker panel was an independent predictor of individual BCR and CR. Prospective studies are needed to further validate this biomarker technology and to define BCR and CR associations in real-world cohorts.

## 1. Introduction

The current mainstay of clinical pathology assessment in prostate cancer is the Gleason grading of hematoxylin and eosin (H&E)-stained tissue sections [1,2,3,4]. Gleason grades are used to stratify patients based on Gleason/International Society of Urological Pathology (ISUP) grade group definitions, which guide treatment decisions and expectations on disease courses for patients [1,2,3,4]. While Gleason/ISUP grade grouping using H&E is validated for predicting biochemical recurrence (BCR) post-surgery, it is well recognised that this method is far from optimal (concordance index < 0.008) [1,5,6,7]. The interpretation of Gleason grading using H&E is highly subjective, resulting in high inter-observer variability and subsequent sub-optimal treatment selection for many individual patients [2,8,9,10,11]. To be able to improve the outcomes of patients diagnosed with prostate cancer, there is a need to transform current pathological risk stratification methodologies [12].

While there have been many attempts to improve the performance of H&E-informed ISUP grading (e.g., using novel risk stratification tools and innovative nomogram/machine learning techniques), there has been little progress in improving clinical prediction performance [13,14,15]. Fundamentally, H&E staining lacks sufficient detail to facilitate accurate interpretation of the complex pathologies underlying prostate cancer [4,16,17,18]. Molecular markers have also been investigated, including prognostic tools Oncotype DX, Prolaris, Promark, and Decipher; however, these are limited by the tumour heterogeneity and have yet to produce substantial additions to current diagnostic stratification systems [19]. To pursue diagnostic improvements that extend beyond the marginal, new staining or visualisation techniques must be developed [8,9,11,20], ideally based on alterations in cell biology that align with different grades of prostate cancer.

Endosomes and lysosomes are directly involved in energy metabolism, cell division, intracellular signalling and cancer pathogenesis [20,21,22,23,24]; unsurprisingly, they have led to the identification of ideal target biomarkers in cancer cells (i.e., biomarkers from this organelle system may enable the visualisation of features that cannot be observed with H&E stains) [25,26,27,28,29,30,31,32,33]. Appl1, Sortilin and Syndecan-1 are biomarkers within the endosome-lysosome system, and immunohistochemistry (IHC) labelling of these proteins may enable improved visualisation of the complex pathologies underlying prostate cancer [20]. The latter study demonstrated improvement in the subjectivity of grading, but these biomarkers still require evaluation for their capacity to reliably facilitate risk stratification within the ISUP grade groups.

This study aimed to compare the BCR and clinical recurrence (CR) prediction performance of ISUP grade groups defined using H&E-stained slides versus slides that were IHC-labelled for Appl1, Sortilin and Syndecan-1.

## 2. Materials and Methods

### 2.1. Patient Cohort

This study was a retrospective assessment of men with prostate cancer who had consented to provide radical prostatectomy tissue block samples to the Adelaide node of the Australian Prostate Cancer Bioresource (APCB) between January 2006 and August 2014. At the time of radical prostatectomy, these patients were treatment-naïve and had a prostate-specific antigen (PSA) level above 0.2 ng/mL. Tissue block samples were formalin-fixed paraffin-embedded (FFPE) at the time of collection.

For each patient within the cohort, clinical follow-up on BCR and metastatic disease progression/CR was available for a minimum of 10 years post-surgery. At the time of tissue block sampling, data were available on statuses of extracapsular extension (yes vs. no), surgical margin (positive versus negative), seminal vesicle involvement (yes vs. no), perineural involvement (yes vs. no) and lymphovascular involvement (yes vs. no). Data on cribriform and intraductal carcinoma status of the prostate were not available and were deemed inappropriate to generate on the single FFPE tissue block examined (sample bias).

Ethics approval for the present study was obtained through the institutional review board of the University of South Australia (Application IDs: 201907 and 36070).

### 2.2. ISUP Grading According to H&E and IHC Methodologies

For each patient, a representative FFPE tissue block, as determined by the bioresource, was cut at 2 µm to obtain four serial sections. These representative blocks were identified by a uropathologist associated with the APCB.

The first section was stained according to routine H&E methodologies [34]. The section was provided to an independent board of 11 international genitourinary pathologists who came to a consensus and assigned an ISUP grade group to each patient.

According to the technique described by Martini et al. [20], the remaining three serial sections were IHC-labelled with Appl1, Sortilin and Syndecan-1, respectively. The labelling was detected using a DAKO EnVision™ + System (Dako Australia Pty Ltd., West Gosford, NSW, Australia), as previously described [20,24]. Briefly, benign glands were identified with basal cell labelling of Appl1 and Syndecan-1, with minimal labelling occurring in their adjacent secretory epithelium. Well-formed gland morphologies were highlighted by intense perinuclear Sortilin labelling, and poorly-formed gland morphologies by intense diffuse Syndecan-1 labelling. Well-formed glands (Sortilin labelled) were assigned Gleason pattern 3, while poorly-formed glands (Syndecan-1 labelled) were assigned Gleason pattern 4 or above (considering other morphological characteristics including cribriform or fused glands for pattern 4 and sheets, cords, single cells, solid nests and necrosis for pattern 5). Gleason patterns were used to derive ISUP grade groups, which in turn allowed risk stratification of patient tissue samples. A representative micrograph of the immunolabelling used to assign Gleason Pattern 3 and Pattern 4 is illustrated in Appendix A.

Blinded and separated by at least 1 week from the H&E assessment (limiting any potential for memory bias), the same independent board of 11 international genitourinary pathologists examined the three individual IHC-labelled slides and, on consensus, assigned an ISUP grade group to each patient.

### 2.3. Outcomes

The prediction of CR was the primary outcome, and BCR was assessed as a secondary outcome. CR was defined as the time interval from the day of radical prostatectomy to the day of clinical disease progression. CR was documented by the treating clinicians as radiographic metastatic disease development. Patients were censored from the analysis on the last day of follow-up if CR had not been observed. BCR was defined as the time interval from the day of radical prostatectomy to either the day of two consecutive PSA elevations above 0.2 ng/mL, to the initiation of radiation therapy or to androgen deprivation therapy subsequent to PSA elevation that remained below 0.2 ng/mL. Patients were censored from the analysis on the last day of follow-up if a PSA > 0.2 ng/mL had never been reached.

### 2.4. Statistical Analysis

All statistical analyses were performed in R (version 4.1.0). The association between (1) ISUP grade groups derived from the H&E slides and (2) ISUP grade groups derived from the Appl1, Sortilin and Syndecan-1 IHC-labelled slides with both BCR and CR were modelled using Cox proportional hazard approaches. Akaike information criterion (AIC) and visual checks were used to evaluate potential non-linear effects between ISUP grade groups and outcomes. Associations were reported as hazard ratios (HR) with 95% confidence intervals (95% CI). Statistical significance was set at a threshold of *p* < 0.05. The predictive performance of the ISUP grade groups derived from the H&E slides was compared to the ISUP grade groups derived from the Appl1, Sortilin and Syndecan-1 IHC-labelled slides using the concordance statistic (C-statistic). The C-statistic is a statistical measure of the predictive accuracy of a time-to-event regression model [35]. It is similar to the area under the Receiver Operating Characteristic curve used for binary outcomes [36]. Analyses adjusted for extracapsular extension, positive surgical margin, seminal vesicle involvement, peri-neural involvement and lymphovascular involvement were conducted. These adjusted analyses were conducted to evaluate the prognostic independence of the H&E and IHC-informed ISUP grade groups from known prognostic factors. Kaplan–Meier analysis was used to visualise BCR and CR estimates according to H&E- and IHC-informed ISUP grade groups.

## 3. Results

### 3.1. Patient Cohort

The available cohort included 114 patients with prostate cancer, whose tissue samples were ISUP graded using an H&E slide and then independently graded using Appl1, Sortilin and Syndecan-1 IHC-labelled slides. Appendix A provide a description of the patient characteristics and concordance between ISUP grade groups within the cohort. In short, 66 patients (58%) received the same ISUP grade group with the use of the IHC-labelled slides as compared to the H&E slide. Opposingly, 34 patients (30%) had an increase, and 14 patients (12%) had a decrease in their ISUP grade group with the use of the IHC-labelled slides, as compared to the H&E slide.

Patients ranged in age from 45–75 years old at radical prostatectomy and had a BMI between 22 and 39. For pre-radical prostatectomy, most patients had a PSA level between 4.5 ng/mL and 22 ng/mL, while a small percentage (7.9%) were lower than 4.5 ng/mL. Patient staging, as defined by the TNM staging system, ranged from 2A to 3C (please see Martini et al. [20] for additional details). The median [95% CI] follow-up in the cohort was 130 (120–144) months.

### 3.2. Prognostic Significance of ISUP Grade Groups with Clinical Outcomes

As defined by model fit, the association between H&E- and IHC-assisted ISUP grade groups with BCR and CR was best described by a linear relationship (i.e., the HR representing the association between ISUP grade groups is for a 1 unit increase in the ISUP grade group). Table 1 presents the univariable and adjusted Cox analysis results of the association between H&E- and IHC-assisted ISUP grade groups with BCR and CR. On univariable analysis, H&E ISUP grade groups were identified as significantly associated with both CR (HR [95% CI]; 1.6 [1.2–2.1]; *p* = 0.002) and BCR (1.4 [1.1–1.7]; *p* = 0.001). Similarly, on univariable analysis, it was identified that the IHC-assisted ISUP grade groups were significantly associated with both CR (2.0 [1.4–2.8]; *p* < 0.001) and BCR (1.6 [1.3–1.9]; *p* < 0.001). Of the two methods, IHC-assisted ISUP grade groups were demonstrated to be more predictive than H&E ISUP grade groups for both CR (C-statistic = 0.71 versus 0.66) and BCR (C-statistic = 0.63 versus 0.59) within the cohort. Furthermore, on adjusted analysis, the IHC-assisted ISUP grade groups were demonstrated as independently associated with both CR (HR [95% CI]; 1.8 [1.2–2.8]; *p* = 0.009) and BCR (1.4 [1.1–1.8]; *p* = 0.02), which was not observed for the H&E ISUP grade groups for either CR (1.3 [0.9–2.0]; *p* = 0.1) or BCR (1.2 [1.0–1.5]; *p* = 0.1). Figure 1 presents Kaplan–Meier observations of the time to CR and BCR according to the defined H&E- and IHC-assisted ISUP grade groups.

### 3.3. IHC-Assisted ISUP Grade Group Reclassifications

The most common up-classifications to occur in the patient cohort were within the 47 patients graded by H&E, initially classified as ISUP grade group 1; on evaluation of the IHC slides, 15 of these patients were reclassified to ISUP grade group 2, four to ISUP grade group 3 and one to ISUP grade group 4 (one patient was down-classified to benign). Notably, patients that were up-classified had meaningfully higher risk of both CR and BCR. The 10-year incidence of CR was 8% in those who remained classified as ISUP grade group 1, which then increased to 22% in those who were up-classified. Similarly, the 10-year incidence of BCR increased from 39% to 61%, respectively. Figure 2 presents Kaplan–Meier observations of the time to CR and BCR according to IHC-assisted up-classifications in patients H&E-graded as ISUP 1. These observations outline that patients up-classified trended towards worse CR and BCR outcomes compared to their counterparts who remained in ISUP grade group 1—a larger cohort than herein would be required to statistically evaluate this observation.

The most common down-classifications to occur in the patient cohort were within the 27 patients assigned to an ISUP grade group of 2 by H&E; on evaluation of the IHC slides, six were reclassified to ISUP grade group 1 (four patients were up-classified to ISUP grade group 3). Within this small subgroup, a trend in the validity of the IHC-assisted down-classifications was observed within the 10-year incidence of CR, which was 28% in those who remained classified as ISUP grade group 2, and that decreased to 0% in those who were down-classified. Similarly, the 10-year incidence of BCR was observed to decrease from 41% to 33%. Figure 3 presents Kaplan–Meier observations of the time to CR and BCR according to IHC-assisted down-classifications in patients H&E-graded as ISUP grade group 2.

## 4. Discussion

The current study demonstrated that IHC-assisted grading using Appl1, Sortilin and Syndecan-1 biomarkers improved prognostic predictions in patients with prostate cancer compared to H&E. The biomarker panel provided information independent of other common prognostic factors, indicating a more accurate reporting of the pathology. The study illustrates the potential power of the biomarker technology to assist in the contemporary interpretation of Gleason/ISUP grading for improved prediction of BCR and CR, which would be an important step towards a novel precision medicine solution for patients with prostate cancer.

Several clinical models are available to predict BCR, including the D’Amico risk stratification scheme, the Cancer of the Prostate Risk Assessment (CAPRA) score and nomograms from the Memorial Sloan Kettering Cancer Centre (MSKCC) [13,37,38,39,40,41]. These nomograms are limited by their heavy reliance on H&E interpretation for Gleason grading, which has underlying problems with observer variability and confounding pathology. Further investigation is required to assess the performance of IHC-assisted Appl1, Sortilin and Syndecan-1 grading compared to the D’Amico, CAPRA and MSKCC risk stratification schemes. In addition, the likelihood of BCR is largely based on ISUP grade group averages, which do not give an accurate indication for an individual. Nearly half of the patients graded as ISUP 1 by H&E were upgraded by the biomarker panel assessment, and this resulted in predictions with fewer ISUP 1 patients experiencing BCR. This change in grading has the potential to transform clinical practice by utilising biomarker-informed biology to report more accurately on cancer pathology [20].

There are limited tools available for reliable prediction of CR in clinical practice. The most frequently used methods are the tumour node metastasis (TNM) system and the MSKCC nomogram [40,41,42,43]. These tools have demonstrated significant limitations in reliably stratifying patient risks for CR [40,41]. In this study, Appl1, Sortilin and Syndecan-1 demonstrated significant potential to assist improved CR predictions compared to H&E. For example, of the 47 patients assigned to ISUP grade group 1 by H&E, 15 patients were reclassified to ISUP grade group 2 after evaluation of the slides by Appl1, Sortilin and Syndecan-1 IHC—four to ISUP grade group 3 and one to ISUP grade group 4 (one patient was down-classified to benign). Notably, the 10-year incidence of CR was 8% in those who remained classified as ISUP grade group 1, which increased to 22% in those who were up-classified. This provides an indication that the reclassifications appeared appropriate and supported the statistical observation of a higher CR prediction performance (C-statistic) for IHC-assisted ISUP grading versus H&E grading.

The largest proportions of IHC-assisted reclassifications occurred in the upgrading of patients from the H&E-based ISUP grade group 1 and the downgrading of patients from the H&E-based ISUP grade group 2. These reclassifications appear appropriate due to improved CR and BCR prediction performances for IHC-assisted ISUP grading versus H&E grading. This has implications for discriminating between Gleason patterns 3 (ISUP 1) and 4 (ISUP ≥ 2), which is critical for predicting patient outcomes. On adjusted analysis, the biomarker-assisted discrimination of Gleason patterns was associated with an improved prediction of patient outcome, independent from known clinicopathological prognostic variables. This ability of the biomarkers to discriminate between the two Gleason patterns is therefore central to the improved risk class stratification.

## 5. Conclusions

In summary, in a small retrospective cohort study, a biomarker panel based on Appl1, Sortilin and Syndecan-1 was a strong independent predictor for both CR and BCR outcomes, with a higher prediction performance (C-statistic) than H&E-based pathological grading. Preliminary evidence suggested that ISUP grade group reclassifications (both up and down) based on the biomarker panel (as opposed to H&E grading) resulted in better risk stratification of patients. Additional studies should be conducted to investigate the effects of androgen sensitivity and insensitivity on patient stratification when using the biomarker panel. Prospective cohort studies are now being conducted to validate the biomarker technology, and there should be further considerations to investigate the findings in larger real-world cohorts, including for long-term survival outcomes.

## 6. Patents

Funding for this project was provided by Envision Sciences Pty Ltd., the University of South Australia, a Cancer Council SA Beat Cancer Grant, the Movember Foundation/Prostate Cancer Foundation of Australia’s Research Program and the Australian Federal Government (NHMRC development grant GNT1092904 and MTP Connect Biomedical Translation Bridge Program grant BTBR200074). L.M.B. was supported by a Beat Cancer SA Beat Cancer Project Principal Cancer Research Fellowship (PRF1117). A.M.H is supported by an Emerging Leader Investigator Grant from the National Health and Medical Research Council, Australia (APP2008119).

D.A.B. is from the University of South Australia, I.R.D.J. is also from the University of South Australia and L.M.B. is from the University of Adelaide and hold patent WO2014197937A1 ‘Methods for Detecting Prostate Cancer’, which involves this manuscript (Original Patent). The Original Patent holders have appointed UniSA Ventures Pty Ltd., the wholly-owned commercialisation arm of the University of South Australia, to manage the commercialisation of the Original Patent. UniSA Ventures Pty Ltd. has entered into arm’s-length arrangements with Envision Sciences Pty Ltd., under which UniSA Ventures will receive financial benefits from the successful commercialisation of the Original Patent.

Envision Sciences Pty Ltd. is a privately-owned Australian company that is commercialising its work in the field of cancer diagnostics and holds additional patents, including PCT/AU2020/050925 “Methods for Confirming Detection and Evaluating the Progression of a Prostate Cancer” involving the invention reported in this manuscript and is using this with the Original Patent under licence. Envision Sciences Pty Ltd. has engaged the University of South Australia on arm’s-length terms to conduct research and development work, including the subject matter in this manuscript.

D.A.B. is a professor and leader of the Mechanisms in Cell Biology and Disease Research Group in Clinical and Health Sciences at the University of South Australia. D.A.B. is a founding shareholder and benefits from Envision Sciences Pty Ltd.’s research funding. J.J.O. is a shareholder and benefits from Envision Sciences Pty Ltd.’s research funding.

D.A.B., L.K., J.M.L., C.M., A.E., A.S., B.S.-Y.U. and S.P. are employees of the University of South Australia, L.M.B. is an employee of Adelaide University and J.J.O. is an employee of Trinity College Dublin, and each receive benefit from the funding provided by Envision Sciences Pty Ltd., the University of South Australia and the Australian Federal Government for their research work.

## Figures and Tables

**Figure 1 cancers-15-03215-f001:**
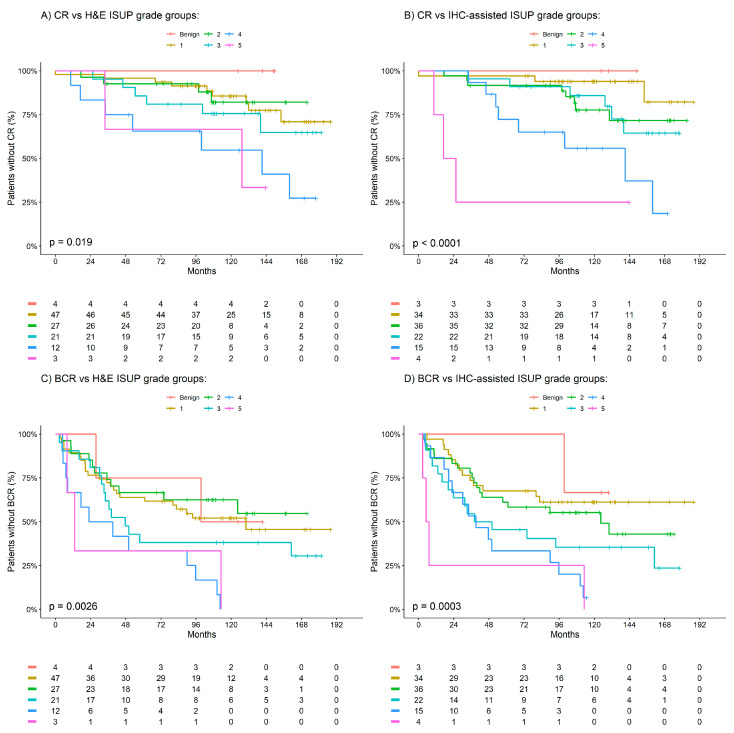
Kaplan–Meier estimates of clinical recurrence (CR) and biochemical recurrence (BCR) by H&E and IHC ISUP grade groups. (**A**) CR vs. H&E ISUP grade groups. (**B**) CR vs. IHC-assisted ISUP grade groups. (**C**) BCR vs. H&E ISUP grade groups. (**D**) CR vs. IHC-assisted ISUP grade groups. Benign patients are indicated in red, ISUP 1 in yellow, ISUP 2 in green, ISUP 3 in teal, ISUP 4 in blue and ISUP 5 in pink.

**Figure 2 cancers-15-03215-f002:**
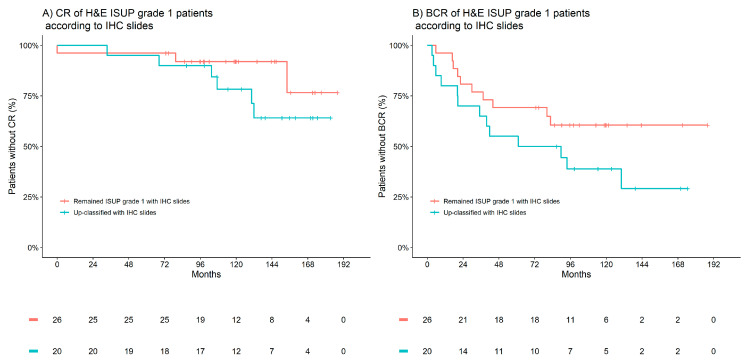
Kaplan–Meier estimates of clinical recurrence (CR) and biochemical recurrence (BCR) according to IHC-associated up-classifications in patients H&E-graded as ISUP 1. (**A**) CR of H&E ISUP grade 1 patients according to IHC slides. (**B**) BCR of H&E ISUP grade 1 patients according to IHC slides. Red indicates patients who remained ISUP grade 1 with IHC, and blue indicates patients up-classified with IHC.

**Figure 3 cancers-15-03215-f003:**
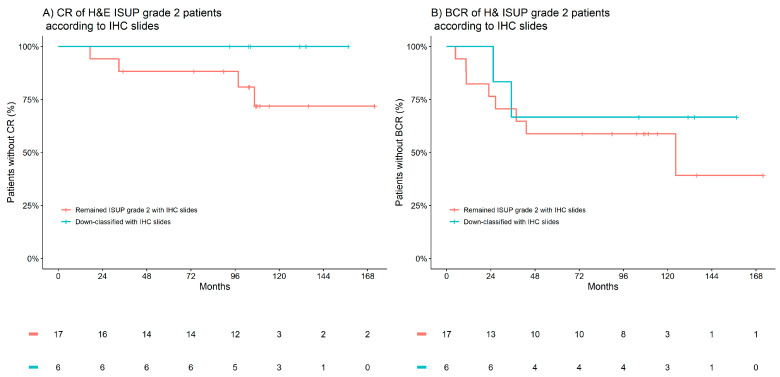
Kaplan–Meier estimates of clinical recurrence (CR) and biochemical recurrence (BCR) according to IHC-associated down-classifications in patients H&E-graded as ISUP 2. (**A**) CR of H&E ISUP grade 2 patients according to IHC slides. (**B**) BCR of H&E ISUP grade 2 patients according to IHC slides. Red indicates patients who remained ISUP grade 2 with IHC, and blue indicates patients down-classified with IHC.

**Table 1 cancers-15-03215-t001:** Association of H&E and IHC-assisted ISUP grade groups with clinical recurrence and biochemical recurrence.

		Univariable			Adjusted ^#^	
	n	HR * [95% CI]	*p*	c	HR * [95% CI]	*p*
Clinical recurrence						
H&E ISUP grade groups	114	1.6 [1.2–2.1]	0.002	0.66	1.3 [0.9–2.0]	0.1
IHC-assisted ISUP grade groups	114	2.0 [1.4–2.7]	<0.001	0.71	1.8 [1.2–2.7]	0.009
Biochemical recurrence						
H&E ISUP grade groups	114	1.4 [1.1–1.7]	0.001	0.59	1.2 [1.0–1.5]	0.1
IHC-assisted ISUP grade groups	114	1.6 [1.3–1.9]	<0.001	0.63	1.4 [1.1–1.8]	0.02

* Hazard ratio (HR) with 95% confidence interval (95% CI) for a 1 unit increase in the ISUP grade. ^#^ Analyses adjusted for extracapsular extensions, surgical margins, seminal vesicle involvements, perineural involvements and lymphovascular involvements at the time of tissue block sampling.

## Data Availability

Data are available for bona fide researchers who request it from the authors.

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
