# Peer review of "Prediction of Prostate Cancer Biochemical and Clinical Recurrence Is Improved by IHC-Assisted Grading Using Appl1, Sortilin and Syndecan-1"

_cancers, 2023, doi:10.3390/cancers15123215_

Round 1
Reviewer 1 Report
Logan et al., reveal an improved method to enhance prediction of PCa biochemical recurrence and clinical recurrence with IHC-assisted grading using Appl1, Sortilin, and Syndecan-1. The study is a nice pilot study to show initial proof of concept for this new biomarker tool for enhanced PCa grading. It is a nicely written manuscript but could be enhanced by additional analyses and clarifications. Please consider addressing the following concerns.
Major Concerns:
1) The authors do not consider additional endosome and lysosome markers. Why were Appl1, Sortilin, and Syndecan-1 the only three factors considered in this IHC-panel for detection? Do these need to be individually stained or can they be co-stained to enhance efficiency for utility?
2) Is the prognostic outcome difference between H&E and IHC-assisted methods statistically different?
3) How did the IHC-assisted method compare to clinical models (including the D’Amico risk stratification scheme, the Cancer of the Prostate Risk Assessment (CAPRA) score and nomograms from the Memorial Sloan Kettering Cancer Center (MSKCC)) to predict BCR?
4) Was the IHC-assisted method compared to an additional cohort? What is the impact of this panel on racial disparity?
Minor Concerns:
1) Please define ISUP in the abstract on line 39.
2) Please list the additional “morphological characteristics” mentioned on line 125.
3) Please add one patient example of Appl1, Sortilin, and Syndecan-1 IHC staining to visualize the intensity utilized.
4) Bold the significant values in Table 1A.
5) Please add stats to all panels in Figures 1-3. Are these statistically relevant?
6) Add details in figure legends for each panel for Figs 1-3, not just the title of the figure.
Reviewer 2 Report
The article entitled ‘Prediction of prostate cancer biochemical and clinical recurrence is improved by IHC-assisted grading using Appl1, Sortilin and Syndecan-1’ is very interesting and provides important evidence suggesting that Appl1, Sortilin and Syndecan-1 are an important biomarker for prostate cancer. However, the authors could improve the article.
· Please include age group and other demographic information of the patients used in the study. A table description is recommended.
· Can the authors provide representative images of staining for the biomarkers used in the study.
· The authors described the staining of the biomarkers to different Gleason grades of prostate cancer tissue sections in section 2.2, Can the authors corelate the biomarker expression levels based on Gleason score to various stages of prostate cancer?
· Can the authors describe if they investigated other biomarkers such androgen or TGFbeta expression in the tissues to understand if it is androgen dependent or androgen independent prostate cancer, to corelate the current biomaker panel(Appl1, Sortilin, Syndecan-1) .
· Can the authors speculate on the mechanistic aspect such as signal transduction in this context.
· The article by Song et al, ‘APPL proteins promote TGFβ-induced nuclear transport of the TGFβ type I receptor intracellular domain describes the role of APPL in Prostate cancer’ is relevant to the current research. It is worth referencing the article.
Reviewer 3 Report
Jessica et al studied a the application of novel biomarker panel (Appl1, Sortilin and Syndecan-1) as a strong independent predictor for both clinical and biochemical recurrence outcomes. The study showing outstanding merits and clinical advantages to help better grading and tumor assessment. The authors aimed to compare the BCR and clinical recurrence (CR) prediction performance of ISUP grade groups defined using H&E-stained slides versus slides that were IHC labelled for Appl1, Sortilin and Syndecan-1.
The only once concern is about the small cohort if it could be expanded to larger population.
Minor issues
Please add some representative H&E as well IHC results
Overall, the study have sufficient merits and novality.
Reviewer 4 Report
Cancers
Topical Collection "Biomarkers for Detection and Prognosis of Prostate Cancer"
Dr. Sanjay Gupta
Guest Editor
Dear Editor,
I have reviewed the study carried out by Logan J, et al., entitled: “Prediction of prostate cancer biochemical and clinical recurrence is improved by IHC-assisted grading using Appl1, Sortilin and Syndecan-1”. This retrospective study evaluates the usefulness of three antibodies (Appl1, Sortilin and Syndecan-1) by immunohistochemical method, in cases of prostate adenocarcinoma from prostatectomy specimens, and their relationship in both clinical and biochemical recurrence. In brief, the authors analyzed the immunohistochemical expression of Appl1, Sortilin and Syndecan-1 in 114 patients diagnosed with prostate cancer, comparing this immunohistochemical expression with the ISUP histopathological classification. In addition, data such as clinical follow-up of biochemical recurrence and metastatic disease progression were obtained. Also, extracapsular extension, surgical margin, seminal vesicle involvement, perineural and lymphovascular involvement, were available. The authors conclude that IHC-assisted staging was more predictive than H&E for both biochemical and clinical relapse. Additionally, IHC-assisted ISUP classification using the biomarker panel was an independent predictor of individual biochemical and clinical relapse. This is an interesting topic for readers of the journal. Here are my comments:
Materials and Methods:
Line 95: radical prostatectomy tissue block….
Did the authors use any inclusion criteria or criteria for the selection of patients? For example present tumor volume, dominant nodule, etc?. It would be advisable to add inclusion and exclusion criteria.
Line 116: According to the technique described by Martini el al [19]….
I have reviewed the reference indicated by the authors and it describes the expression of the three antibodies studied. Reference Figure 3 demonstrates the expression patterns in benign cases, prostatic intraepithelial neoplasia, prostatic intraductal carcinoma, and adenocarcinoma, with similar expression patterns. Given these findings, and taking into account that the immunohistochemical expression can be influenced by different pre-analytical factors, as well as by interpretation, it would be convenient to add a more adequate explanation of how the immunohistochemical assessment was carried out in the cases analyzed. It is necessary to apply a quantitative method that includes the number of positive cells and the intensity of the reaction. Likewise, in cases with tumor heterogeneity, how was the reality obtained with the ISUP grade group? Was a cutoff point identified? An extensive explanation of immunohistochemical expression aspects is necessary.
Results:
Line 166: Table A1 and Table 2.
These tables should be identified as an appendix or supplementary material, since it creates confusion with line 177, Table A1.
Line 173: Prognostic significance of ISUP grade groups with clinical outcomes
The authors analyze the ISUP grade group and the immunohistochemical study. However, there are a number of variables that can influence these results and are not analyzed (almost 70% of the patients had stage pT3). Is there any additional relationship with the stage?
Line 191: IHC-assisted ISUP grade group reclassification.
The authors state that GG1 was the one that showed the highest reclassification through IHC, with a total of 20 up-classified cases, showing BCR and CR comparable to the newly assigned groups. It would be interesting to know what the pT stage was in these cases.
In the graphs shown (Figure 1 C and D) it is impressive that two clearly defined groups could be differentiated, especially in the one that identifies BCR Vs IHC assisted. Is there a significant difference at 24, 48 and 72 months?
Discussion
It would be advisable to include a part of the limitations of the study since it is a retrospective study.
Minor comments:
Line 38: please remove point (.)
Line 214: please remove 3.4 Figures, Tables and Schemes
The references must be modified according to the journal's requirement.
Reviewer 5 Report
This is a well written manuscript which addresses an important clinical topic. My questions and concerns are listed below:
1. Abstract: I think it would be helpful to note that the patient samples were from prostatectomies and to also note median time to BCR and CR.
2. Introduction, line 62: I think it would be helpful to note the C-statistic reported in these (or other) studies – this would help provide a point of reference for the reader.
3. Introduction: Please consider noting the potential role of other molecular markers in predicting disease progression, for example perhaps some of the molecules noted in this review article: https://www.ncbi.nlm.nih.gov/pmc/articles/PMC9385485/. Also please comment on how comparative utility of OncotypeDX, Prolaris, Promark, and Decipher.
4. Introduction, lines 77-85: It is unclear to me why using IHC that specifically looks at expression of molecules associated with endosomes and lysosomes would be more beneficial compared to using IHC that looks at expression of molecules elsewhere in the cell that are known to be associated with cancer progression. Please explain.
5. Results section 3.1: I do not see tables A1 or A2 (patient characteristics) in the manuscript – are these supplemental files? I think these should be included in the main text.
